# An Investigation of the Neurotoxic Effects of Malathion, Chlorpyrifos, and Paraquat to Different Brain Regions

**DOI:** 10.3390/brainsci12080975

**Published:** 2022-07-24

**Authors:** Ekramy Elmorsy, Ayat Al-Ghafari, Huda Al Doghaither, Mohamed Salama, Wayne G. Carter

**Affiliations:** 1Department of Forensic Medicine and Clinical Toxicology, Faculty of Medicine, Mansoura University, Mansoura 35516, Egypt; ekramyelmorsy@mans.edu.eg (E.E.); mohamed.salama@gbhi.org (M.S.); 2Pathology Department, Faculty of Medicine, Northern Border University, Arar 91431, Saudi Arabia; 3School of Medicine, University of Nottingham, Royal Derby Hospital Centre, Derby DE22 3DT, UK; 4Department of Biochemistry, Faculty of Science, King Abdulaziz University, Jeddah 21589, Saudi Arabia; abalghafari@kau.edu.sa (A.A.-G.); haldoghaither@kau.edu.sa (H.A.D.); 5Scientific Research Center, Dar Al-Hekma University, Jeddah 22246, Saudi Arabia; 6Cancer and Mutagenesis Unit, King Fahd Medical Research Centre, King Abdulaziz University, Jeddah 22252, Saudi Arabia; 7Institute of Global Health and Human Ecology, The American University in Cairo (AUC), Cairo 11385, Egypt

**Keywords:** acetylcholinesterase, chlorpyrifos, malathion, neurodegenerative diseases, neuropathy target esterase, neurotoxicity, paraquat, pesticides, redox stress

## Abstract

Acute or chronic exposures to pesticides have been linked to neurotoxicity and the potential development of neurodegenerative diseases (NDDs). This study aimed to consider the neurotoxicity of three widely utilized pesticides: malathion, chlorpyrifos, and paraquat within the hippocampus (HC), corpus striatum (CS), cerebellum (CER), and cerebral cortex (CC). Neurotoxicity was evaluated at relatively low, medium, and high pesticide dosages. All pesticides inhibited acetylcholinesterase (AChE) and neuropathy target esterase (NTE) in each of the brain regions, but esterase inhibition was greatest in the HC and CS. Each of the pesticides also induced greater disruption to cellular bioenergetics within the HC and CS, and this was monitored via inhibition of mitochondrial complex enzymes I and II, reduced ATP levels, and increased lactate production. Similarly, the HC and CS were more vulnerable to redox stress, with greater inhibition of the antioxidant enzymes catalase and superoxide dismutase and increased lipid peroxidation. All pesticides induced the production of nuclear Nrf2 in a dose-dependent manner. Collectively, these results show that pesticides disrupt cellular bioenergetics and that the HC and CS are more susceptible to pesticide effects than the CER and CC.

## 1. Introduction

Pesticides are widely utilized industrial and residential chemicals and primarily include herbicides, insecticides, fungicides, and rodenticides. Further sub-classification of pesticides can be made based on their chemical structures; for example, organophosphates (OPs) represent the commonly employed chemical subset of insecticides, estimated at approximately 30% of the global insecticide market and approximately 13% of the European Union market [1].

Malathion (MAL) is a broad-spectrum OP insecticide widely used in agriculture as well as public health pest control programs, such as mosquito eradication [2]. MAL is considered probably carcinogenic, listed as a Class III (slightly hazardous) chemical [3], and can be metabolized in the liver to malaoxon (MO), an approximately 100-fold more potent and essentially irreversible inhibitor of acetylcholinesterase (AChE), with mammalian IC30 and IC50 values of approximately 0.1 and 2.4 µM, respectively [4,5]. MAL modulates the expression and activity of a number of enzymes and proteins that have a role in OP-induced neurobehavioural effects and developmental neurotoxicity after acute or chronic exposures, and these act through cholinergic and non-cholinergic mechanisms, including induction of cellular oxidative stress, neuroinflammation, and apoptosis [2,6,7,8,9].

Chlorpyrifos (CPF), like MAL, is one of the most widely utilized OPs for agricultural purposes and is considered a moderately hazardous (Class II) pesticide [3]. Bioactivation of CPF produces chlorpyrifos-oxon (CPO), a metabolite that is also a potent and near irreversible AChE inhibitor with an IC30 of approximately 30 nM [4,10,11]. A link has been proposed between acute and subacute CPF exposure and the induction of neurological, developmental, and neurobehavioural anomalies in adults and children, and this may relate to non-cholinergic mechanisms including induction of redox stress [11,12].

Paraquat (PQ) is a commonly employed herbicide with potentially serious neurotoxic effects after either acute or chronic exposures. PQ is structurally similar to the neurotoxic chemical MPP+ (1-methyl-4-phenylpyridinium), a metabolite of N-methyl-4-phenyl-1,2,3,6-tetrahydropyridine (MPTP). MPTP crosses the blood–brain barrier (BBB) and undergoes biotransformation by astrocytic monoamine oxidase-B (MOA-B) to 1-methyl-4-phenyl-2,3-dihydropyridinium (MPDP) and then undergoes further oxidation to MPP+, which is taken up by neurons through dopamine transporters (DAT).

MPP+ can damage mitochondria, inhibit mitochondrial complex I activity, and induce cellular redox stress [13,14,15]. Sufficient exposure to MPP+ induces Parkinsonian phenotypes in humans as well as experimental animals [16,17,18], but a lack of typical environmental exposure to this agent renders it an unlikely source of idiopathic Parkinson’s disease (PD). Hence, although PQ is structurally similar to MPP+, it has a low affinity for DATs, and therefore its ability to act as a toxin to specifically trigger the loss of dopaminergic neurons and induce Parkinsonian phenotypes remains equivocal. However, PQ can damage mitochondria and induce the production of reactive oxygen species (ROS) and associated redox stress, a common pathological mechanism detected in NDDs including Alzheimer’s disease (AD) and PD [19,20,21,22].

The acute inhibition of AChE by MAL, CPF or PQ can lead to excessive ACh accumulation and overstimulation of postsynaptic ACh receptors in neurons or muscle to trigger a cholinergic toxidrome [23]. However, non-cholinergic mechanisms can also contribute to neurotoxicity via off-target binding effects [23,24,25,26,27]. Indeed, OP pesticides can induce delayed neurotoxicity in humans, termed OP-induced delayed neurotoxicity (OPIDN) which is characterized by axonal degeneration and is triggered by OP inhibition of neuropathy target esterase (NTE) [27,28].

Collectively, exposure to exogenous chemicals such as environmental or domestic pesticides including OPs and PQ has been proposed as a risk factor for certain NDDs [20,23,27,29,30,31,32,33,34].

Patients with Alzheimer’s disease (AD), the most common NDD, experience clinical sequelae that include progressive cognitive decline and episodic memory loss, consistent with neuronal loss and damage to the frontotemporal lobes, including the hippocampal and entorhinal cortex regions [35,36,37].

PD is the most common NDD with motor-related symptomology, typically characterized by a clinical triad of bradykinesia, postural rigidity, and resting tremor, as well as a range of non-motor symptoms including sleep and psychiatric disorders [38]. Damage and loss of dopaminergic neurons of the substantia nigra pars compacta (SNpc) within (and that innervate) the basal ganglia of the corpus striatum is a defining pathological hallmark of PD [38,39,40].

Although the etiology of many NDDs remains incompletely understood, the common findings of mitochondrial damage and oxidative stress suggest that they have a crucial role as contributing factors to disease pathogenesis and/or progression. Hence agents, such as pesticides, that elicit these undesired cellular responses may contribute to neurodegeneration by a common molecular mechanism. However, if acute exposure to a specific pesticide had a more prominent role in one particular NDD, then it might be expected that more specific and localized brain damage may arise. The aim of the present study was to consider the neurotoxicity of three commonly employed pesticides, MAL, CPF, and PQ and to examine if there were differential effects between these pesticides on cellular bioenergetics and oxidative stress within the hippocampus (HC), corpus striatum (CS), cerebellum (CER), and cerebral cortex (CC) of male albino rats. Relative neurotoxicity of these pesticides was also considered via their effects on the common target enzymes, AChE and NTE.

## 2. Materials and Methods

### 2.1. Chemicals

All chemicals were purchased from Sigma (St. Louis, MO, USA) unless specified otherwise. Malathion (MAL) (O,O-dimethyl-S-1,2-bis ethoxy carbonyl ethyl phosphorodithionate) and chlorpyrifos (CPF) (O,O-diethyl O-[3,5,6,-trichloro-2-pyridyl] phosphorothionate) were dissolved in dimethylsulphoxide (DMSO), while paraquat (PQ) (1,1-dimethyl-4,4′-bipyridinium) was dissolved in sterile saline. ATP and lactate assays kits were obtained from Abcam (Cambridge, MA, USA).

### 2.2. Animal Protocols

This study was approved by the local bioethics committee of Dar Al-Hekma University Scientific Research Center (Study reference; RC/2021/001). Three month old male albino rats (Wistar strain, 195 ± 25 g body weight) were used and housed in groups of six rats at 22 ± 3 °C with access to water and food. A total of 66 rats were acclimatized for one week before experimentation and subdivided into groups of 6 rats. Three groups received MAL, one group each received MAL at a dose of 50, 100, or 150 mg/kg. Similarly, three groups received CPF, one group each received CPF at a dose of 25, 50, or 100 mg/kg. Three groups each received PQ at a dose of 1, 5, or 10 mg/kg. One group received DMSO as vehicle control for the OP pesticide experiments (MAL and CPF treatments), and one group received sterile saline as vehicle control for the PQ treated group. Pesticides, DMSO or saline were administered intramuscularly (IM) to the thigh quadriceps muscles each day for 21 days for the MAL, CPF, and DMSO (control) groups, while PQ (or saline control) was administered as a single dose. Pesticide dosages were estimated from previous experimental data conducted in our laboratory that induced toxicity measurements and were below the lethal dose for half of the animals (LD50) by this (results not included) and other routes of administration [41,42,43]. Rats were monitored for signs of acute toxicity including drowsiness, lethargy, and locomotor changes, as well as production of diarrhea and changes in pupil size, food intake, and/or body weight. Rats that displayed any signs of acute toxicity during the course of the studies were administered atropine sulfate (0.05 mg/kg as required). Atropine usage was required for two rats after MAL administration at the high (150 mg/kg) dosing and for one rat after high CPF (100 mg/kg) dosing. At 24 h after the last dose of MAL and CPF, and 7 days after the single PQ injection, rats were sacrificed under thiopental anesthesia by decapitation (refer to Appendix A). Brains were collected and chilled on ice-filled culture plates and then dissected to separate the hippocampus (HC), corpus striatum (CS), cerebral cortex (CC), and cerebellum (CER). Brain sections were washed in saline and then stored at −80 °C until required.

### 2.3. Acetylcholinesterase Assays

Brain regions were homogenized on ice using a glass hand-held homogenizer in a 10% (*w*/*v*) buffer of 10 mM Tris/HCl pH 8 [25]. Cholinesterase activity was quantified based on the method of Ellman et al. (1961) [44] modified for a microtiter plate. In a 96-well plate, 75 µL of 50 mM Tris-HCl buffer (pH 8) was added to 50 µL of brain tissue homogenate from the HC, CS, CER, or CS brain regions. Then, 3 mM dithio-2-nitrobenzoic acid (DTNB) solution (containing 20 mM magnesium chloride (MgCl_2_) and 100 mM NaCl) was added, and plates were incubated at 37 °C for 20 min before the addition of 0.001 M acetylcholine iodide (ATCI) substrate to start the reaction. The production of 2-nitro-5-thiobenzoate was read at 412 nm at 1-min time intervals for a total of 3 min using a plate reader (‘TopCount’ (Perkin Elmer, Ueberlingen, Germany). A substrate-free blank was run in parallel with each sample tested. Experiments were conducted at least three times for each pesticide concentration.

### 2.4. Neuropathy Target Esterase (NTE) Assays

Brain NTE activity was quantified in the HC, CS, CER, and CS brain regions following the method of Johnson (1977) [45]. Briefly, 50 μL of tissue homogenate from each brain region was added to 1.95 mL of 40 mM paraoxon. Tubes containing a blank solution composed of paraoxon and 50 mM mipafox were run in parallel with each test sample. Tubes were incubated at 37 °C for 20 min. Afterwards, 2 mL of a dispersion solution containing 500 mM phenylvalerate and 0.03% (*v*/*v*) Triton X-100 was added. After 15 min, 2 mL of 1% (*w*/*v*) sodium dodecyl sulfate (SDS) in 0.025% 4-aminoantipyrine buffer was added to stop the reaction. Finally, 0.5 mL of 0.8% (*w*/*v*) potassium ferricyanide was added and the colour from the 4-N-(1,4-benzoquinoneimine)antipyrine chromophore was read at 512 nm using a spectrophotometer (DU 800 spectrophotometer, Beckman Coulter, Carlsbad, CA, USA).

### 2.5. Mitochondrial Complex I and III Assays

Brain regions were homogenized using a glass hand-held homogenizer in SETH buffer (1:10 (*w*/*v*), pH 7.4 (250 mM sucrose, 2 mM ethylenediamine tetraacetic acid (EDTA), 10 mM Trizma base, 50 IU/mL heparin)). Homogenates were centrifuged at 800× *g* for 10 min at 4 °C to pellet cell debris, and then supernatants were removed and flash-frozen in liquid nitrogen and stored at −80 °C until required for mitochondrial enzyme activity determination. Protein concentration was determined by the Bradford assay [46], using bovine serum albumin (BSA) as a protein standard.

For complex I assay, homogenates were subjected to three cycles of freeze–thawing to disrupt the mitochondrial membranes and maximize complex I enzymatic recovery. Assays for complexes I and III were conducted following the methods of Spinazzi et al. (2012) [47]. Briefly, 10 µg of protein homogenate from each brain region was mixed with 700 µL of distilled water in a 1 mL cuvette. A total of 100 µL of potassium phosphate buffer (0.5 M, pH 7.5), 60 µL of fatty acid–free BSA (50 mg/mL), 30 µL of KCN (10 mM), and 10 µL of NADH (10 mM) were added to the cuvette. After mixing, the basal absorbance of the cuvette was read in a spectrophotometer at 340 nm (DU 800 spectrophotometer, Beckman Coulter, Carlsbad, CA, USA) for 2 min. Then, 6 µL of ubiquinone 1 (10 mM) was added to start the reaction, and the absorbance was read (at 340 nm) for 2 min. In parallel, another cuvette was similarly processed but prepared with the addition of 10 µL of 1 mM rotenone (as a specific complex I inhibitor).

For complex III assays, 5 µg of brain homogenate was added to a 1 mL cuvette containing 730 µL of distilled water, 50 µL of potassium phosphate buffer (0.5 M, pH 7.5), 75 µL of oxidized cytochrome c, 50 µL of KCN (10 mM), 20 µL of EDTA (5 mM, pH 7.5), and 10 µL of Tween-20 (2.5% (*v*/*v*)). The final volume was adjusted to 990 µL via the addition of distilled water and then baseline absorbance read at 550 nm for 2 min before initiation of the reaction by the addition of 10 µL of 10 mM decylubiquinol and the absorbance read (at 550 nm) for a further 2 min. In parallel, another cuvette was similarly processed but with the addition of 10 µL of 1 mg/mL antimycin A (as a specific complex III inhibitor).

For both mitochondrial complex assays, the background was corrected by subtraction of the absorbance of samples without lysate. Specific complex I or III activity for each sample was calculated as the difference between the cuvette readings in the absence and presence of rotenone or antimycin A, respectively. Specific complex enzyme activity (nmol min^−1^ mg^−1^) = (∆ Absorbance/min × 1000)/[(extinction coefficient × volume of sample used in mL) × (sample protein concentration in mg mL^−1^)]. Experiments were conducted at least 4 times for each concentration of a brain sample. The extinction coefficient was estimated as 6.2 mM^−1^ cm^−1^ for NADH, for complex I specific activity measurements. A value of 18.5 mM^−1^ cm^−1^ for reduced cytochrome c was used for mitochondrial complex III measurements [47]. Experiments were conducted at least 5 times for each brain region and each neurotoxin exposure concentration.

### 2.6. ATP Assays

A total of 10 mg of the different brain region samples was homogenized in perchloric acid (PCA). Excess PCA was then precipitated by the addition of ice-cold 2M KOH (15% of the total homogenate volume), and then the homogenate was neutralized with Tris base (pH = 7). ATP levels were then quantified using an ATP assay kit (ab83355, Abcam, Cambridge, UK) following the manufacturer’s protocol. The assay is based on the phosphorylation of glycerol to generate a product quantified by a colourimetric method. Absorbance was read at 570 nm via the spectrophotometer (DU 800 spectrophotometer, Beckman Coulter, Carlsbad, CA, USA), and ATP content was expressed as µmoles/mg of sample homogenate.

### 2.7. Lactate Assays

Quantitation of lactate was performed using a spectrophotometric L-lactate assay kit according to the manufacturer’s guidelines (Biovision, Mountain View, CA, USA). In brief, 10 mg of brain tissue was washed in ice-cold saline and then homogenized in 400 µL of assay buffer. The homogenate was centrifuged for 5 min at 14,000 rpm at 4 °C to remove pelleted insoluble material. The supernatant was collected and kept on ice. Deproteinization was performed with PCA before samples were centrifuged at 13,000× *g* for 15 min at 4 °C, and the supernatant was retained. A dilution factor for deproteinization was calculated for each sample; 25 µL of each sample was pipetted into a 96-well plate and incubated with the reaction mixture at room temperature for 30 min. Results were then measured at 450 nm via a spectrophotometer (DU 800 spectrophotometer, Beckman Coulter, Carlsbad, CA, USA). The absorbance readings from blank wells were subtracted from samples and standards, and a standard curve was plotted. Readings from brain samples were interpolated from the standard curve, and the final concentration of lactate was calculated. Lactate levels were normalized to the protein concentrations. Experiments were performed in triplicates.

### 2.8. Catalase Activity Assay

Catalase (CAT) was assayed colourimetrically at 620 nm and was expressed as µmoles of H_2_O_2_ consumed per minute per mg of protein (U^b^) according to the method described by Singh et al. (2008) [48]. Each reaction mixture contained 1.0 mL of 0.01 M pH 7 phosphate buffer, 0.1 mL of tissue homogenate, and 0.4 mL of 2 M H2O2. The reaction was stopped by the addition of 2 mL of dichromate-acetic acid reagent.

### 2.9. Superoxide Dismutase Activity Assay

Assays for superoxide dismutase-1 (SOD-1) were based on the SOD-mediated inhibition of the reduction of nitroblue tetrazolium to blue formazan by superoxide anions, as described by Beauchamp and Fridovich (1971) [49]. Units of SOD activity were calculated and expressed in terms of mg of total protein.

### 2.10. Quantitation of Lipid Peroxidation Products

Lipid peroxidation was estimated in brain tissues by quantifying the levels of thiobarbituric acid reactive substance (TBARS), according to the method of Niehaus and Samuelson (1968) [50]. In brief, the supernatant from tissue homogenates was treated with tertiary butanol-trichloroacetic acid-hydrochloric acid reagent, mixed thoroughly, and then kept in a boiling water bath for 15 min. After cooling, the tubes were centrifuged for 10 min, and the supernatant used for spectrophotometric measurement at 535 nm using a spectrophotometer (DU 800 spectrophotometer, Beckman Coulter, Carlsbad, CA, USA) and expressed as mmoles TBARS per 100 g tissue.

### 2.11. Nuclear Factor E2-Related Factor 2 (Nrf2) Quantitation

Nrf2 levels were quantified in nuclear extracts from brain samples according to the manufacturer’s protocol using a commercial kit (Nrf2 Transcription Factor Assay Kit, Abcam, Cambridge, UK). Briefly, 40 μL of complete binding buffer (CBB) was added to each well, and then 10 μL of each sample or a blank was added to the wells of a 96-well plate containing an immobilized double stranded DNA consensus binding site for Nrf2. The plate was incubated for 1 h at room temperature on a rocking platform at 100 rpm. Wells were washed 3 times with 200 μL 1× wash buffer, and then immunodetection of bound protein was initiated by incubation with a primary antibody at 1:1000 dilution for 1 h at room temperature. The plate wells were washed before incubation with secondary antibody (1:1000 dilution) for 1 h at room temperature. The plate wells were then washed before the addition of a developing solution (100 μL/well). After 10 min, a stop solution was added, and the absorbance of the wells read at 450 nm (DU 800 spectrophotometer, Beckman Coulter, Carlsbad, CA, USA). Blank values were subtracted from test sample absorbances.

### 2.12. Statistical Analyses

Groups of data were compared using either one-way or two-way ANOVAs with Bonferroni post-tests to compare the effects of the tested pesticides at the three dosing concentrations and for the four different brain regions for enzymatic activities, bioenergetics measurements, and redox status. Statistical analyses were performed using GraphPad Prism 5.0 (GraphPad Software Inc., San Diego, CA, USA). Statistical significance was defined as a *p* value of <0.05. The effect of neurotoxins on mitochondrial complex enzyme kinetics was evaluated using nonlinear regression. A MichaelisMenten model was used to determine enzymatic Ki (inhibitor concentration at half-maximal inhibition) and Vmax (maximum enzyme velocity).

## 3. Results

### 3.1. Pesticides Differentially Inhibit Brain AChE and NTE

Although the pesticides MAL, CPF, and PQ decreased AChE activity in a dose-dependent manner, neither the endogenous activity of AChE (controls) nor the response to pesticides was uniform across the brain regions studied (HC, CS, CER, CC) (Figure 1A–C). Endogenous AChE activity was in the descending order CS > CER > HC > CC. After administration of the lowest dose of MAL (50 mg/kg/day), AChE was significantly inhibited in the HC and CS regions and not the CER or CC. Intermediate dosing with MAL also reduced AChE in both regions as well as the CER. The CC had the lowest endogenous AChE activity and was also the most resistant to MAL exposure, with only the highest dose administered able to significantly reduce AChE activity by 26%.

A similar profile of brain regional responses was observed after dosing with CPF: with only HC and CS regions vulnerable to AChE inhibition at the lower dosing regimens and AChE significantly inhibited within the CC only at the highest dosing treatment. PQ dosing followed the same pattern except that the intermediate dosing also induced significant inhibition of AChE within the CC region (Figure 1A–C).

Endogenous NTE activity was relatively uniform across the brain regions, but the vulnerability of the HC and CS brain regions to the effects of pesticides was evidenced by the significant inhibition of NTE by all three pesticides and at all of the pesticide doses (Figure 2A–C). By comparison, low pesticide dosing for either MAL, CPF, or PQ was ineffectual for inhibition of NTE within either the CER or CC brain regions (Figure 2A–C). Furthermore, the levels of inhibition of NTE at high pesticide dosing were 28–52% for the HC, 49–58% for the CS, but only 25–32% for the CER, and 20–25% for the CC. A two-way ANOVA confirmed that the pesticide inhibition of AChE or NTE reflected pesticide dose and brain region (Appendix A).

### 3.2. Pesticides Differentially Inhibit Cellular Bioenergetics

The impact of pesticides on cellular bioenergetics was examined via activity assay measurements of mitochondrial complex I (MC I) and mitochondrial complex III (MC III) from within each of the brain regions. Pesticide dosing reduced MC I and MC III activities in a dose-dependent manner (Figure 3A–F). Endogenous mitochondrial activities were highest in the CC and then CER brain regions. MC I activity was not significantly inhibited in any of the brain regions after low dosing but was significantly inhibited at the intermediate and high dosing regimens.

MC III was more sensitive to inhibition by each of the pesticides, and similar to the serine hydrolase measurements (Section 3.1), the HC and CS regions experienced the greatest declines in activity, falling by 54–64% (HC) and 55–75% (CS) of basal levels after high pesticide dosing compared with 27–31% (CER) and 36–49% (CC). The resistance of the CER and CC brain regions to pesticide effects was also evidenced by the observation that only high dosing with CPF or PQ triggered a significant reduction of MC III activity. Differences between the effects of the pesticides on MC I and MC III activities were evaluated by a two-way ANOVA, and a Michaelis–Menten model was used to determine values for the enzymatic maximal velocity (Vmax) and inhibitor constant (Ki) (Appendix A). Enzymatic capacity as Vmax and inhibitor potential as Ki were similar across brain regions and for both MC I and MC III, with a notable increase in Ki at the highest pesticide dosing.

Cellular ATP levels declined in a dose-dependent manner in response to pesticide treatments, with significant reductions at either the intermediate or high pesticide dosing (Figure 4A–C). The differential sensitivity to pesticide effects between brain regions was apparent with only high pesticide dosing able to significantly reduce cellular ATP levels in the CER or CC brain regions.

An examination of the production of cellular lactate was used to further evidence altered cellular bioenergetics. Lactate production increased according to the pesticide dosage and mirrored the corresponding reduction of ATP levels such that there was significant lactate production in response to intermediate or high pesticide dosing for the HC and CS brain regions, but only at high pesticide dosing for the CER and CS brain regions (Figure 4D–F).

### 3.3. Pesticides Induce Cellular Redox Status

Pesticides decreased the activities of the antioxidant enzymes, catalase (CAT) (Figure 5A–C) and superoxide dismutase (SOD) (Figure 5D–F) in dose-dependent manners. Akin to the previous results, the HC and CS were relatively more vulnerable to pesticide effects than the CER and CS, with significant reductions of antioxidant enzymatic activities at all pesticide doses. Collectively, reductions in CAT activities were greater than those for SOD, and the reduction of both activities was highest in the HC and CS brain regions, particularly at high dosing. By comparison, SOD activity was reduced at the high pesticide dosing by 22–34% (HC) and 30–42% (CS), but only 16–27% (CER) and 17–26% (CC). Similarly, CAT activity declined by 39–43% (HC) 44–59% (CS) compared with 22–46% (CER) and 31–45% (CC) at the high pesticide dosages.

In keeping with the pesticide-induced redox stress, there was a parallel increase in lipid peroxidation measured as TBARS (Figure 6A–C). The HC and CS brain regions were again more sensitive to pesticide dosing effects, with significant effects even at the lowest dosing regimen, and these regions also displayed the largest increase of TBARS from basal levels, particularly for the CS.

Nuclear activation of the transcription factor Nrf2 promotes induction of cytoprotective proteins in response to cellular stress, and nuclear Nrf2 levels increased according to pesticide dose (Figure 7A–C).

## 4. Discussion

This study was undertaken to examine the neurotoxic effects of three widely utilized pesticides, the OPs MAL and CPF, and the bipyridilium, PQ. These pesticides share common mechanisms of neurotoxicity via inhibition of the serine hydrolase AChE and through induction of redox stress. However, whether these pesticides have altered capacity to inhibit another serine hydrolase, NTE, or whether they differentially affect cellular bioenergetics and elicit cellular damage that is uniform across different brain regions have not previously been considered. The results of this study indicate that for the four brain regions examined, the HC and CS are more susceptible to the toxic effects of these pesticides, including a disruption of cellular bioenergetics, than the CER and CC.

All three pesticides inhibited AChE (Figure 1A–C), the enzyme primarily responsible for limiting the activity of synaptic ACh, and this was observed for all brain regions assessed, albeit with more inhibition within the HC and CS than the CER or CC. Broad effects of the pesticides on the cholinergic system are expected since cholinergic neurotransmission arises from two primary sources of ACh that released from local interneurons such as those of the striatum that also modulate the release of other neurotransmitters, including dopamine [51,52] and those of projection neurons that are located within nuclei that are spread throughout the brain and innervate distal regions of the CNS [51,52].

Inhibition of target AChE is the desired mechanism of action of OP pesticides and, for the relatively inert thions, MAL and CPF, is primarily driven after their bioactivation to oxon forms that are essentially irreversible AChE inhibitors. PQ, although not an OP, is still able to bind and inhibit AChE [53,54]. Collectively, the pesticide inhibition of AChE was mild to moderate, with only the higher pesticide dosing able to induce tissue inhibition levels close to the approximately 50% inhibition considered as a likely threshold for induction of the signs and symptoms of cholinergic toxicity [55]. Similarly, a study of repeated dosing with MAL (15 days, 30 or 100 mg/kg/day) in mice reduced hippocampal AChE by approximately 20 and 45%, respectively, and this was not sufficient to induce signs of cholinergic toxicity [7]. A progressive reduction of AChE activity might be expected due to the repeated pesticide dosing regimen (Appendix A), but this will be offset by rates of spontaneous hydrolysis of the pesticide–AChE complex. The binding of MAL to AChE produces an O,O’-dimethyl group which is less stable (hydrolyzes more rapidly) than the O,O’-diethyl produced after CPF binding, hence cholinergic effects from MAL would be less likely to persist for similar levels of AChE inhibition, assuming parity for the rates of dealkylation (aging). However, the OP-induced cholinergic toxicity in both cases will also be limited due to protein turnover, and mammalian tissue AChE has an estimated half-life of 3–12 days [56]. This may be more rapid for irreversibly inhibited enzymes [56]. At the highest dose of CPF employed (100 mg/kg), acute toxicity was only observed in one rat, but a dose elevation of CPF to 150 mg/kg, at least in hens, is sufficient to induce cholinergic toxicity [57]. Lastly, from molecular modeling and inhibition assays in vitro, PQ presumably forms only electrostatic interactions at the AChE esteratic binding site, hence, relatively weak and presumed transient binding with a 50% inhibition concentration (IC_50_) of approximately 8 µM [54].

Collectively at the doses employed, cholinergic toxicity was minimal, and due to the expected hydrolysis of the bound inhibitor as well as AChE turnover, the impact on ACh signalling (cholinergic mechanisms) would seem unlikely to represent a mechanism associated with long-term health consequences in the dosed animals, although an extended time course of animal study was not undertaken.

Dosing with these pesticides also led to a mild to moderate inhibition of NTE. NTE is a transmembranous protein ubiquitously expressed within tissues including neurons [28,58]. Similar to AChE, NTE is a serine hydrolase and also displays phospholipase and lysophospholipase activities [28]. At sufficient exposures, certain OPs can bind and inhibit NTE, and this has a major role in OPIDN and triggers an axonal degeneration [28,59,60]. However, malathion or malaoxon may only be weak inhibitors of NTE, at least for the hen brain enzyme [28,61], and our pesticide dosing of rats only induced mild to moderate inhibition of NTE and therefore at a level likely to be below that required for NTE-induced neurotoxicity. Indeed, even a repeated CPF dosing regimen for hens did not induce cumulative inhibition of NTE (maximal at ~18%), even with 58–77% concomitant inhibition of brain AChE [62]. Hence, the levels of NTE inhibition detailed herein fall considerably below the approximately 70% inhibition threshold for NTE that along with ageing of the inhibited enzyme, are required to induce the OPIDN that is observed 1–4 weeks after acute or short-term neurotoxic exposures [28,59].

All pesticides resulted in a disruption to the bioenergetic capacity of each brain region with inhibition of mitochondrial complex I and complex III activities (Figure 3A–F). In keeping with our data, dosing with MAL [7], CPF [57], or PQ [63] resulted in reduced mitochondrial complex I activity in other studies. The effect of the three tested pesticides on mitochondrial complex III in experimental animals has been less well studied, but mitochondrial complex III is a major source of hydrogen peroxide production in PQ-induced oxidative stress [19], and CPF administered in vitro (50 and 100 µM) significantly decreased mitochondrial complexes II/III activities in human neuroblastoma SH-SY5Y cells [64]. Collectively, inhibition of mitochondrial complexes I and III will limit the activity of the electron transport chain and the coupling to proton transport across the inner mitochondrial membrane for these two mitochondrial complexes. Hence, there will be a net reduction in the proton motive force that drives ATP production in complex V, and this was consistent with the concomitant reduction in cellular ATP production (Figure 4A–C) and a switch to anaerobic respiration and associated elevation of lactate production (Figure 4D–F).

Increased cellular redox stress is mitigated by the antioxidant enzymes CAT and SOD, as well as cellular thiols such as glutathione. SOD catalyses the dismutation of a superoxide radical to form either molecular oxygen or hydrogen peroxide, and the latter can be further decomposed to water and molecular oxygen via the action of catalase. Consistent with the induction of redox stress and depletion of antioxidant enzyme capacities, pesticide dosing resulted in reduced CAT and SOD activities (Figure 5A–F). Similarly, administration of MAL to rats triggered decreased CAT and SOD activities within the cortex, striatum, cerebellum, and hippocampal brain regions [6], although some MAL dosing regimens also triggered increased CAT and SOD activities [6]. Likewise, relatively high dosing of MAL at 200 mg/kg/day for seven days triggered increased total brain CAT and SOD activity [65], presumably reflecting increased protein expression in response to redox stress.

The pesticide induction of cellular redox stress will be enhanced through mitochondrial damage and liberation of further ROS that can damage proteins, lipids, and DNA. Oxidative damage to lipids was quantified as TBARS, and these were elevated in response to the pesticides, with the greatest levels of lipid peroxidation within the CS at a high dosing (Figure 6A–C). Other independent studies have also reported increased lipid peroxidation in brain tissue in response to dosing with MAL [6,65], CPF [66,67], or PQ [63,68].

In keeping with the stimulation of cellular response to mitigate redox stress, the transcription factor Nrf2 was activated in all brain regions and most notably within the HC and CS (Figure 7A–C). Nrf2 is activated by a number of cellular stimuli that trigger its migration from the cytoplasm to the nucleus to regulate gene transcription including genes involved in antioxidant defense [69,70]. Hence, the increased nuclear Nrf2 is in keeping with a response to cellular redox stress and was most notable with CPF-dosed animals; presumably a reflection of the potency of CPF as a redox stressor.

In summary, although all three pesticides inhibit AChE and NTE, the relatively mild levels of enzymatic inhibition when coupled to rates of dissociation or spontaneous hydrolysis of the pesticide-protein complex as well as protein turnover, indicate that acute and chronic cholinergic toxicity or induction of OPIDN is unlikely at this level of serine hydrolase inhibition. Furthermore, acute OP toxicity will likely decline due to the activity of xenobiotic metabolizing enzymes, including serine and A-esterases for OPs [71]. Hence, the structure–activity relationships of other more sensitive serine hydrolase pesticide targets, such as acylpeptide hydrolase, could account for certain pesticide-induced acute or chronic health effects [24,25,26,72], and these may also be inhibited dissimilarly across different brain regions.

Injury to mitochondria and reduced mitochondrial efficiency and a net reduction of ATP production and concomitant cellular redox stress are common mechanisms of cellular damage from xenobiotic agents, and this includes pesticides. Hence, xenobiotic-induced mitochondrial dyshomeostasis and redox stress are likely a commonality that could contribute to the development and/or progression of NDDs [73]. In this study, the susceptibility to a disruption of cellular bioenergetics was highest within the HC and CS brain regions by comparison to the CER and CC. This could reflect the unequal distribution of pesticides, differential pesticide uptake or regionally specific responses and mitigation of redox stress, such that increased damage was observed in the HC and CS brain regions. Differential brain region responses to pesticides will also be expected based on the dosing, duration of exposure, and time between dosing and tissue analyses [6]. However, irrespective of the mechanisms responsible for the dissimilar response to the pesticides, there are specific areas of the brain that were observed to be relatively vulnerable to damage. This has implications for diseases that display regional histopathologies, such as AD and PD, with damage and loss of functionality of the HC typically observed for AD and damage and the loss of dopaminergic neurons of the SNpc within the basal ganglia of the corpus striatum is a pathological hallmark of PD. However, this study does not provide evidence of causal linkage, and only through more sophisticated and validated methods associated with accurate sampling and quantitation of acute or chronic exposures could a link between pesticide exposures and specific NDDs be established.

## Figures and Tables

**Figure 1 brainsci-12-00975-f001:**
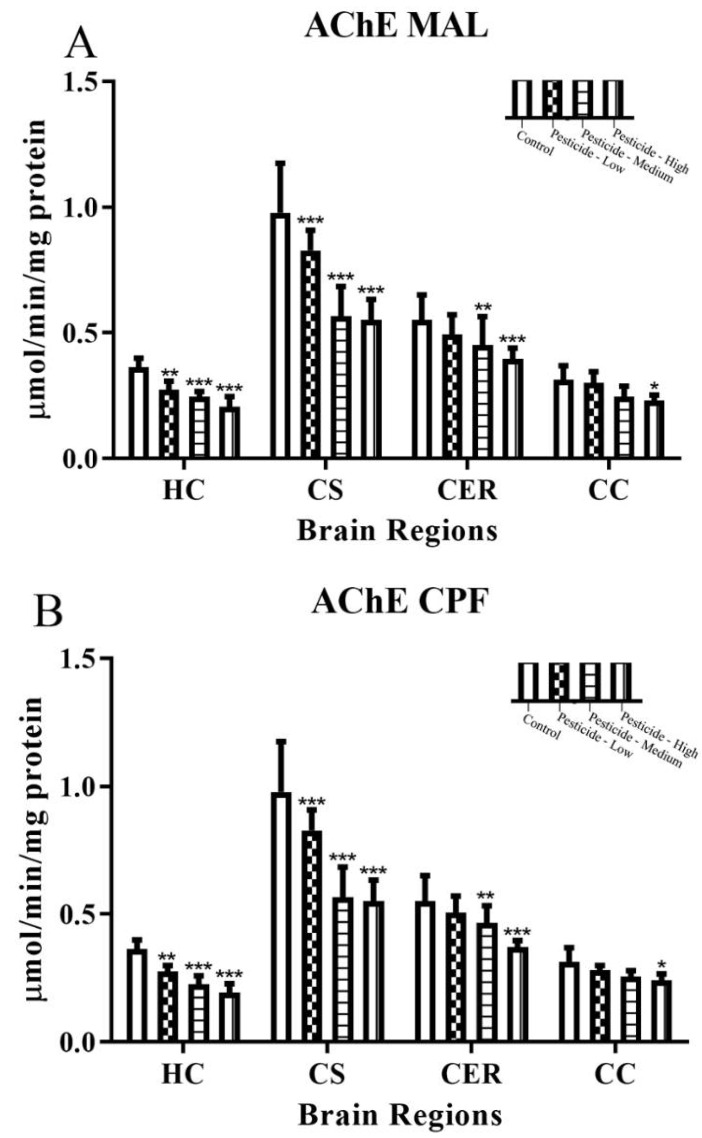
Pesticide-induced inhibition of brain AChE activity. Rats were dosed with malathion (MAL) (**A**); chlorpyrifos (CPF) (**B**); or paraquat (PQ) (**C**); at low, medium, or high dosages and the level of acetylcholinesterase (AChE) activity quantified within the hippocampus (HC), corpus striatum (CS), cerebellum (CER), and cerebral cortex (CC). Histograms are representative of mean values with SDs, with significant changes marked with asterisks, according to * *p* < 0.05, ** *p* < 0.01, and *** *p* < 0.001.

**Figure 2 brainsci-12-00975-f002:**
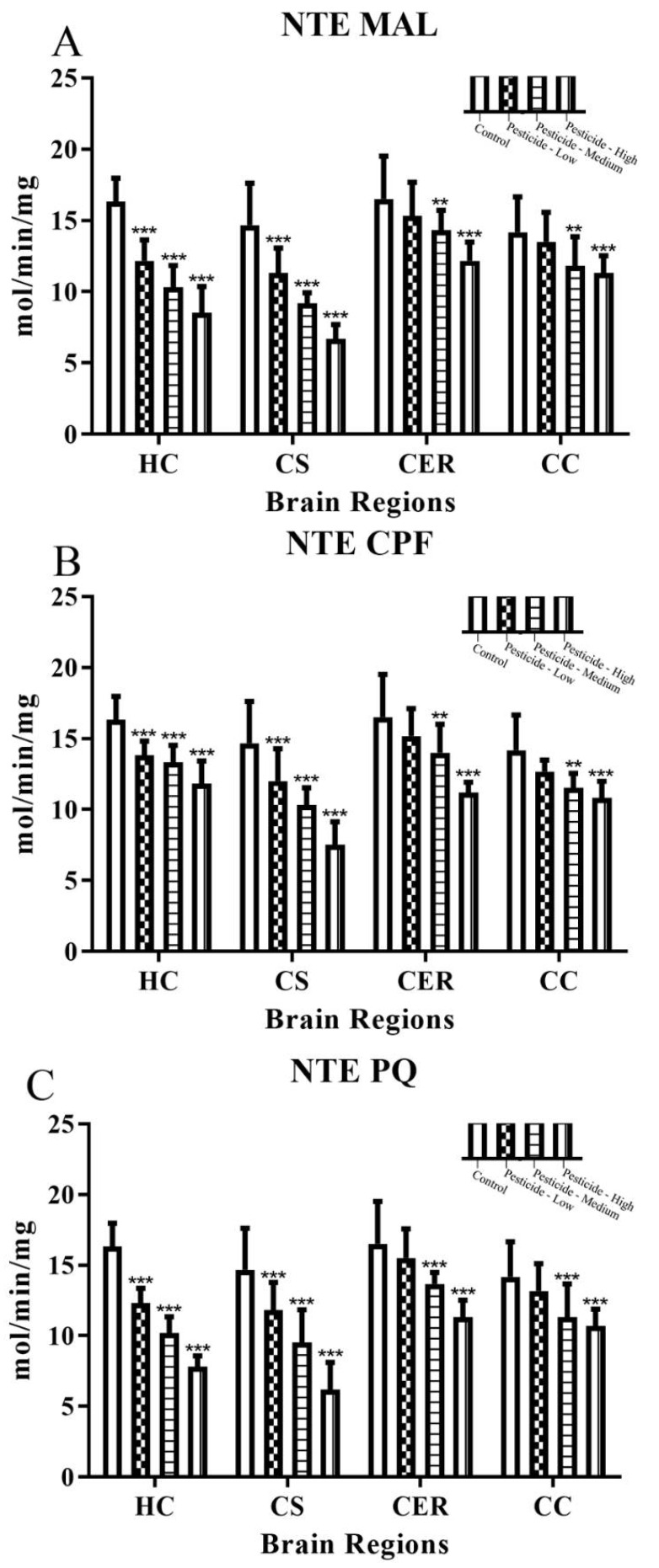
Pesticide-induced inhibition of brain NTE activity. Rats were dosed with malathion (MAL) (**A**); chlorpyrifos (CPF) (**B**); or paraquat (PQ) (**C**); at low, medium, or high dosages and the level of neuropathy target esterase (NTE) activity quantified within the hippocampus (HC), corpus striatum (CS), cerebellum (CER), and cerebral cortex (CC). Histograms are representative of mean values with SDs, with significant changes marked with asterisks, according to ** *p* < 0.01, and *** *p* < 0.001.

**Figure 3 brainsci-12-00975-f003:**
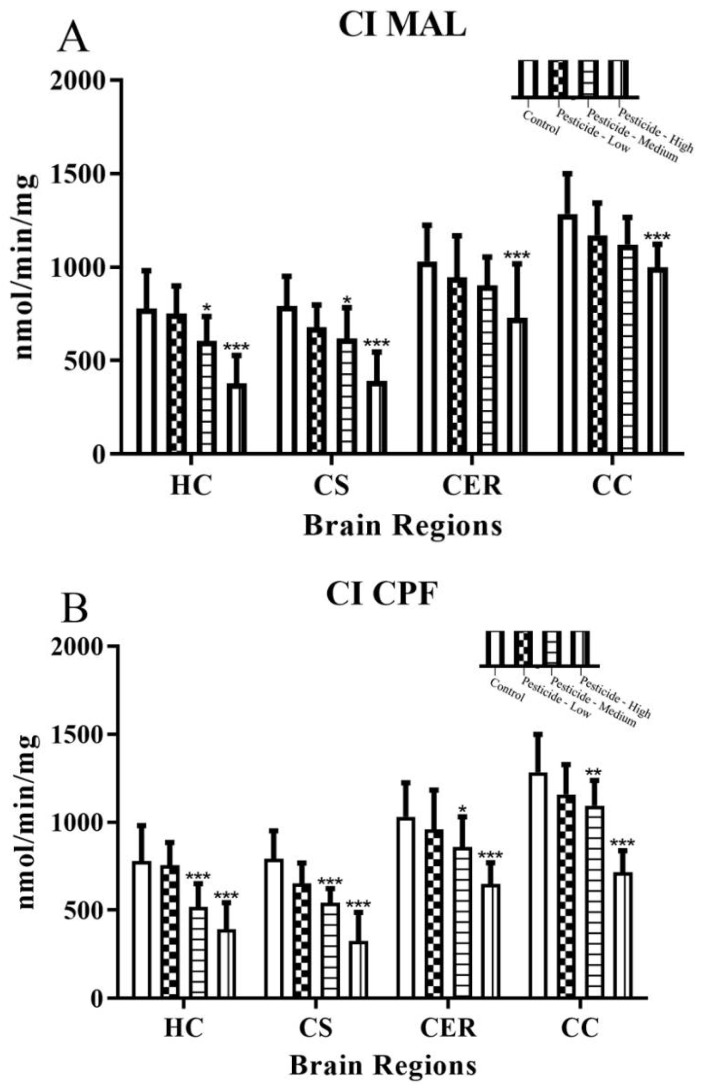
Pesticide-induced inhibition of mitochondrial complex I and complex III activities. Rats were dosed with malathion (MAL) (**A**,**D**); chlorpyrifos (CPF) (**B**,**E**); or paraquat (PQ) (**C**,**F**); at low, medium, or high dosages and the activity of mitochondrial complex I and complex III enzymes quantified within the hippocampus (HC), corpus striatum (CS), cerebellum (CER), and cerebral cortex (CC). Histograms are representative of mean values with SDs, with significant changes marked with asterisks, according to * *p* < 0.05, ** *p* < 0.01, and *** *p* < 0.001.

**Figure 4 brainsci-12-00975-f004:**
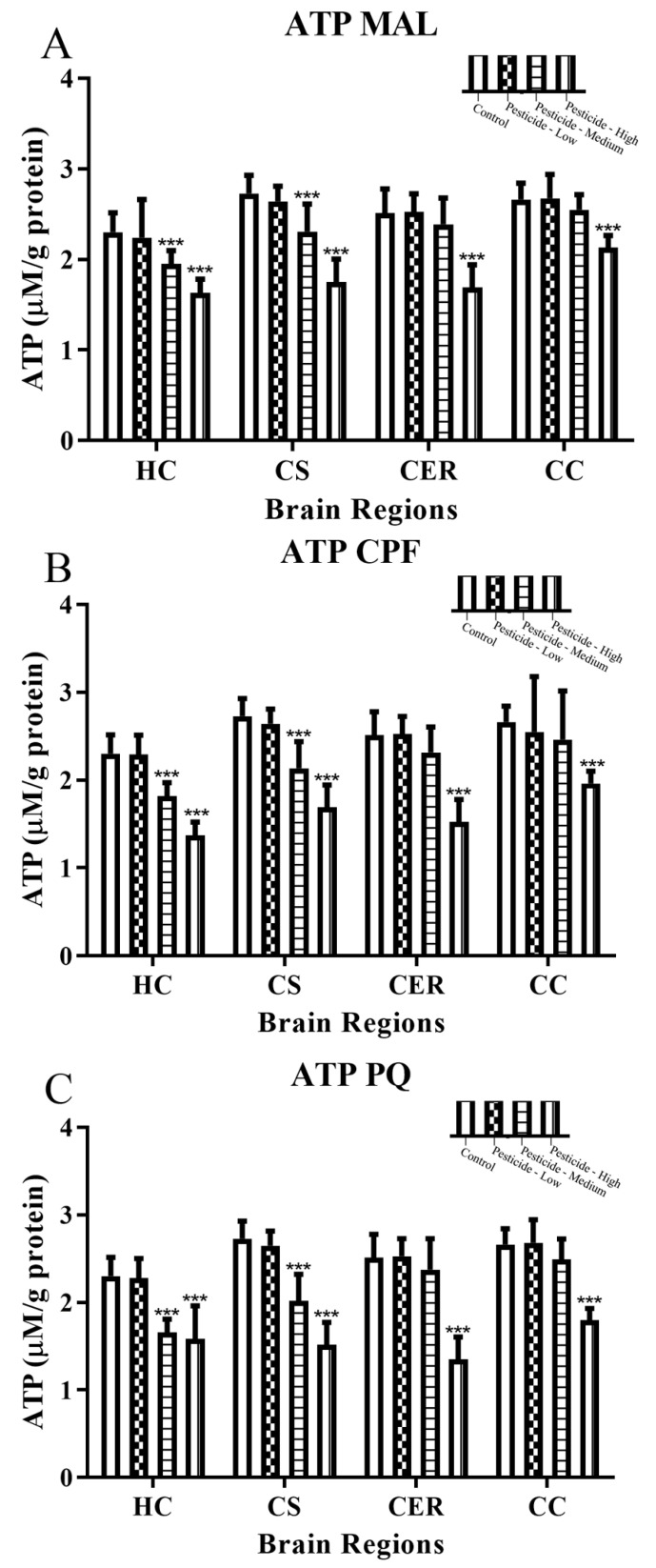
Pesticide-induced inhibition of ATP levels and lactate production. Rats were dosed with malathion (MAL) (**A**,**D**); chlorpyrifos (CPF) (**B**,**E**); or paraquat (PQ) (**C**,**F**) at low, medium, or high dosages and the levels of cellular ATP and lactate quantified within the hippocampus (HC), corpus striatum (CS), cerebellum (CER), and cerebral cortex (CC). Histograms are representative of mean values with SDs, with significant changes marked with asterisks, according to * *p* < 0.05, ** *p* < 0.01, and *** *p* < 0.001.

**Figure 5 brainsci-12-00975-f005:**
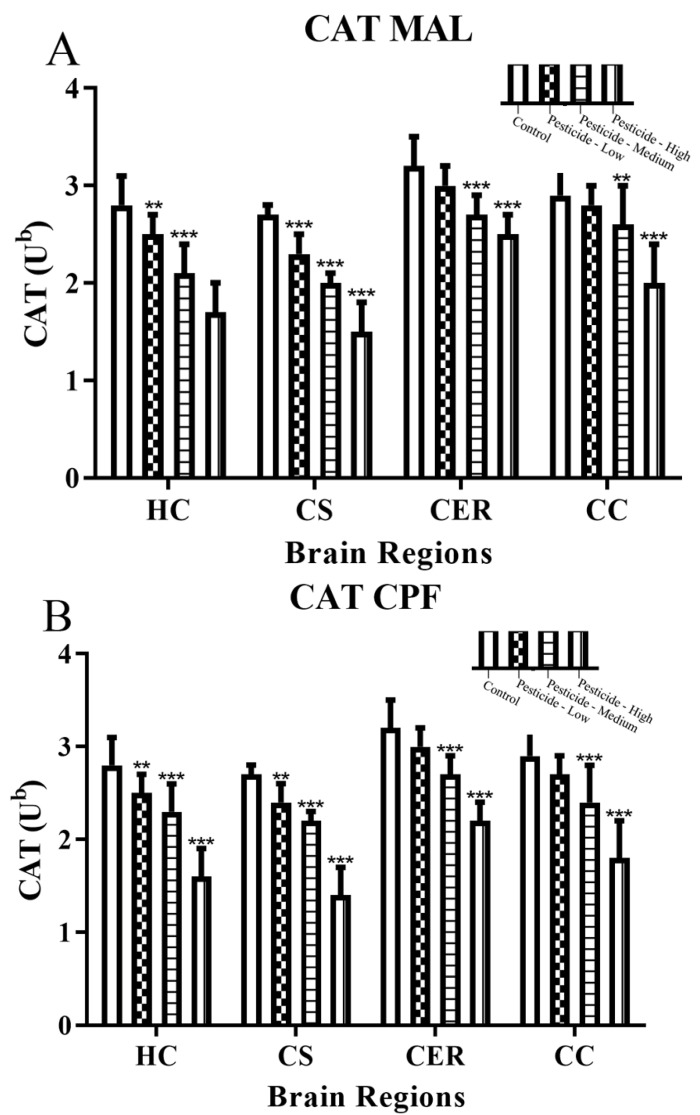
Pesticide-induced inhibition of catalase and superoxide dismutase activities. Rats were dosed with malathion (MAL) (**A**,**D**); chlorpyrifos (CPF) (**B**,**E**); or paraquat (PQ) (**C**,**F**); at low, medium, or high dosages and the activity of catalase (CAT) and superoxide dismutase (SOD) enzymes quantified as µmoles of H_2_O_2_ consumed per minute per mg of protein (U^b^) within the hippocampus (HC), corpus striatum (CS), cerebellum (CER), and cerebral cortex (CC). Histograms are representative of mean values with SDs, with significant changes marked with asterisks, according to ** *p* < 0.01, and *** *p* < 0.001.

**Figure 6 brainsci-12-00975-f006:**
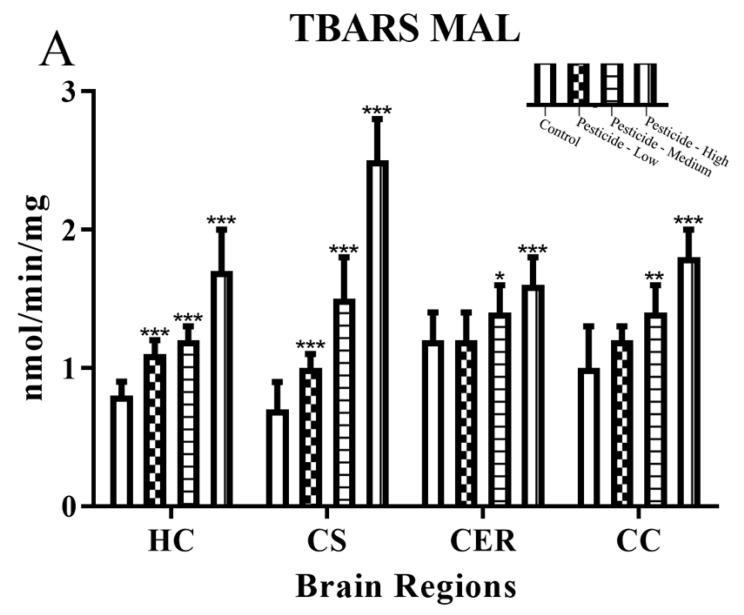
Pesticide induction of thiobarbituric acid reactive substances (TBARS). Rats were dosed with malathion (MAL) (**A**); chlorpyrifos (CPF) (**B**); or paraquat (PQ) (**C**); at low, medium, or high dosages and the levels of thiobarbituric acid reactive substances (TBARS) quantified within the hippocampus (HC), corpus striatum (CS), cerebellum (CER), and cerebral cortex (CC). Histograms are representative of mean values with SDs, with significant changes marked with asterisks, according to * *p* < 0.05, ** *p* < 0.01, and *** *p* < 0.001.

**Figure 7 brainsci-12-00975-f007:**
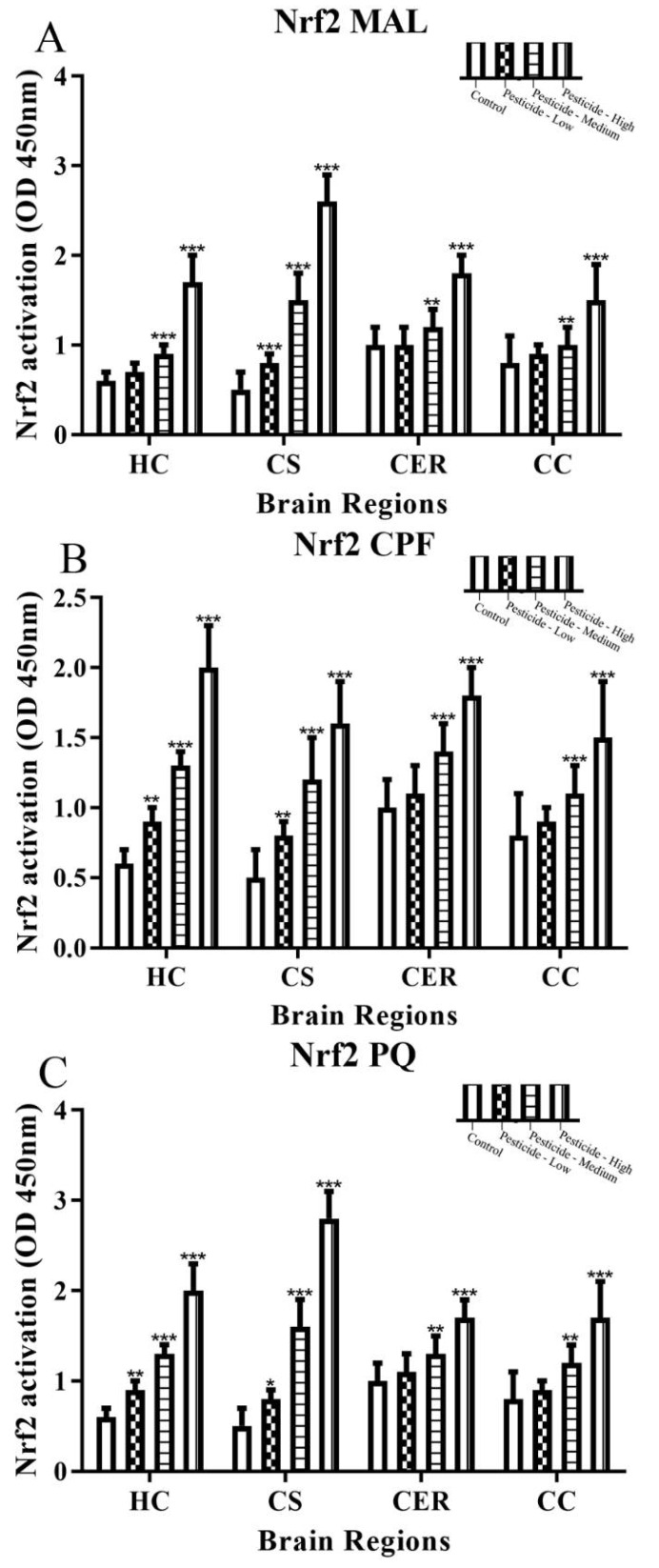
Pesticide induction of Nrf2 levels. Rats were dosed with malathion (MAL) (**A**); chlorpyrifos (CPF) (**B**); or paraquat (PQ) (**C**); at low, medium, or high dosages and the levels of nuclear Nrf2 quantified within the hippocampus (HC), corpus striatum (CS), cerebellum (CER), and cerebral cortex (CC). Histograms are representative of mean values with SDs, with significant changes marked with asterisks, according to * *p* < 0.05, ** *p* < 0.01, and *** *p* < 0.001.

## Data Availability

Data available on request from the first author (Ekramy Elmorsy).

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
