# Peer review of "An Investigation of the Neurotoxic Effects of Malathion, Chlorpyrifos, and Paraquat to Different Brain Regions"

_brainsci, 2022, doi:10.3390/brainsci12080975_

Round 1
Reviewer 1 Report
I have received the paper "The hippocampal and corpus striatum brain regions are relatively vulnerable to pesticide-induced serine hydrolase inhibition, bioenergetic disruption and damage from redox stress" by Elmorsy et al. for review. Authors have shown the neurotoxicity of pesticides: malathion, chlorpyrifos, and paraquat in different brain regions. Authors have provided experimental evidence that these pesticides are inducing neurotoxicity in the brain by disrupting the cellular bioenergetics and antioxidant defense system, especially in the hippocampus and corpus striatum. Although, the study is systematic and well planned however the authors have missed some key experiments in this study. These experiments may help to make the study more impressive and thoughtful. I have some major and minor comments related to this study:
Major comments:
1. Authors should quantify the level of acetylcholine, especially in the hippocampus (HC) and corpus striatum (CS) at a medium dose of each pesticide.
2. Authors have not shown the neurodegenerative effect of aforesaid pesticides on the cholinergic neurons. I would suggest they perform the acetylcholinesterase (AChE) immune-reactivity and quantify the cholinergic neurons (immunohistochemistry) in the most affected regions of the brain at a medium dose of pesticides.
3. Authors have shown that the activities of antioxidant enzymes (catalase and superoxide dismutase) are significantly reduced in HC and CS regions of the brain. Mostly, these pesticides induce neurotoxicity by altering the expression of xenobiotic-metabolizing enzymes and antioxidant defense systems (Singhal et al., 2011; Tiwari et al., 2010; 2012; 2013). However, the study does not provide experimental evidence on whether the disruption of the antioxidant defense system is due to reduced enzymatic activity or because of the altered expression of antioxidant enzymes.
4. Authors claimed that these pesticides share common mechanisms of neurotoxicity. If it’s true, these three pesticides (all three combined) must show a synergistic effect even at subtoxic doses. To support this notion, the authors should perform the experiment to check whether the combination of these pesticides is showing a synergistic effect at subtoxic doses.
Minor comments:
1. The title of the paper should be more concise and impressive?
2. In the discussion section authors have mentioned “HC and CS are relatively vulnerable to the toxic effects of pesticides”. Please cite the appropriate references to support this notion.
3. In the discussion section “This provides a possible link between certain pesticide exposures and impairment of the brain regions typically damaged in NDDs such as Alzheimer’s disease and Parkinson’s disease.” Please cite the appropriate references to support this opinion.
4. The author should review the literature (especially pesticide-induced Alzheimer’s disease and Parkinson’s disease) related to this study and update the introduction and discussion sections.
Author Response
I have received the paper "The hippocampal and corpus striatum brain regions are relatively vulnerable to pesticide-induced serine hydrolase inhibition, bioenergetic disruption and damage from redox stress" by Elmorsy et al. for review. Authors have shown the neurotoxicity of pesticides: malathion, chlorpyrifos, and paraquat in different brain regions. Authors have provided experimental evidence that these pesticides are inducing neurotoxicity in the brain by disrupting the cellular bioenergetics and antioxidant defense system, especially in the hippocampus and corpus striatum. Although, the study is systematic and well planned however the authors have missed some key experiments in this study. These experiments may help to make the study more impressive and thoughtful. I have some major and minor comments related to this study:
Major comments:
- Authors should quantify the level of acetylcholine, especially in the hippocampus (HC) and corpus striatum (CS) at a medium dose of each pesticide.
We thank Reviewer #1 for this useful suggestion. Presumably, Reviewer #1 makes this suggestion as differential effects of pesticides on the cholinergic system could arise in regions of relatively low or high acetylcholine neurotransmitter activity. However, measuring ACh levels directly is challenging and often requires expertise in microdialysis (Nirogi et al. Quantification of acetylcholine, an essential neurotransmitter, in brain microdialysis samples by liquid chromatography mass spectrometry. Biomed Chromatogr. 2010 Jan;24(1):39-48. doi: 10.1002/bmc.1347. PMID: 19877295.) and furthermore, the maintenance of the ACh signal is governed by the activity of AChE, and we do show in Figure 1 that the endogenous levels of AChE do vary between brain regions, with the corpus striatum displaying the highest endogenous activity. Furthermore, we mention in the discussion section that broad effects of the pesticides on the cholinergic system are expected since cholinergic neurotransmission arises from local interneurons such as those of the striatum as well as those of projection neurons located within nuclei spread throughout the brain [references 48,49]. Thus, we appreciate the suggestion but microdialysis methods to confirm localized levels of ACh in the presence and absence of pesticides would constitute a new and full study by itself.
- Authors have not shown the neurodegenerative effect of aforesaid pesticides on the cholinergic neurons. I would suggest they perform the acetylcholinesterase (AChE) immune-reactivity and quantify the cholinergic neurons (immunohistochemistry) in the most affected regions of the brain at a medium dose of pesticides.
We thank Reviewer #1 for this suggestion. The focus of the study was a comparative analysis of the disruption to cellular bioenergetics in response to pesticide exposures and whether this was uniform across brain regions. We appreciate that we may have placed an undue slant on the paper to consider neurodegeneration, simply as a reflection of the literature which tends to make an associated of pesticide exposures with neurodegeneration. A study suggested by Reviewer #1 that specifically considers immune-reactive cholinergic neurons and whether these are damaged by pesticide exposures overlays with the experimentation suggested in point 1, and as such, could form the basis of a new subsequent research study.
- Authors have shown that the activities of antioxidant enzymes (catalase and superoxide dismutase) are significantly reduced in HC and CS regions of the brain. Mostly, these pesticides induce neurotoxicity by altering the expression of xenobiotic-metabolizing enzymes and antioxidant defense systems (Singhal et al., 2011; Tiwari et al., 2010; 2012; 2013). However, the study does not provide experimental evidence on whether the disruption of the antioxidant defense system is due to reduced enzymatic activity or because of the altered expression of antioxidant enzymes.
In Figure 5 (A-C) and Figure 5 (D-F) we do show that the activity of the enzymes CAT and SOD, respectively, are reduced in response to the pesticide exposures and that these reductions in activities are dose-dependent effects. However, Reviewer #1 makes the important point that we do not refer to the activity of xenobiotic-metabolizing (detoxification) enzymes. We have therefore included details regarding the potential influence of xenobiotic-metabolizing enzymes in the discussion section of the manuscript.
- Authors claimed that these pesticides share common mechanisms of neurotoxicity. If it’s true, these three pesticides (all three combined) must show a synergistic effect even at subtoxic doses. To support this notion, the authors should perform the experiment to check whether the combination of these pesticides is showing a synergistic effect at subtoxic doses.
All three pesticides are inducers of redox stress (as shown in our study and those of others (references 6-9 and 12-15 of the manuscript]. In our study, we also show that each of the pesticides has a detrimental effect on cellular bioenergetics. It would be of interest to consider if these three pesticides had either an additive or synergistic effect on bioenergetic disruption and redox stress, but this would involve a separate study that would require specific ethical approval since it would be difficult to limit cholinergic toxicity. This type of experiment may better suit an in vitro approach with neuronal cells, but this also represents a separate discrete study.
Minor comments:
- The title of the paper should be more concise and impressive?
We appreciate the suggestion and have edited the title to make this more focused and representative of the manuscript content.
- In the discussion section authors have mentioned “HC and CS are relatively vulnerable to the toxic effects of pesticides”. Please cite the appropriate references to support this notion.
The relative vulnerability of the hippocampus (HC) and corpus striatum (CS) regions relates to the findings of our study such that that the HC and CS are more susceptible to pesticide effects than the cerebellum (CER) and cerebral cortex (CC). We have amended the text of the abstract and the discussion to make this clear.
- In the discussion section “This provides a possible link between certain pesticide exposures and impairment of the brain regions typically damaged in NDDs such as Alzheimer’s disease and Parkinson’s disease.” Please cite the appropriate references to support this opinion.
This sentence was based on the work of our study that the hippocampus (HC) and corpus striatum (CS) are more vulnerable to pesticide-induced damage than the cerebellum (CER) and cerebral cortex (CC). However, we appreciate that a causal link between pesticide exposures and neurodegenerative disease has not been made in our study. We have therefore deleted this sentence and ensured that the work discussed only covers the actual pesticide effects.
- The author should review the literature (especially pesticide-induced Alzheimer’s disease and Parkinson’s disease) related to this study and update the introduction and discussion sections.
We did perform a literature search in PubMed to cover the latest papers that specifically covered relevant pesticides and neurodegenerative diseases prior to submission of this work. Since submission, there has been one additional relevant paper and we have included that into the revised manuscript: Sule RO, Condon L, Gomes AV. A Common Feature of Pesticides: Oxidative Stress-The Role of Oxidative Stress in Pesticide-Induced Toxicity. Oxid Med Cell Longev. 2022 Jan 19;2022:5563759. doi: 10.1155/2022/5563759.
Reviewer 2 Report
Title : “The hippocampal and corpus striatum brain regions are relatively vulnerable to pesticide-induced serine hydrolase inhibition, bioenergetic disruption and damage from redox stress. “
In this study the authors explored the neurotoxicity of three widely utilized pesticides: malathion, chlorpyrifos, and paraquat, and their potential for differential damage to the hippocampus (HC), corpus striatum (CS), cerebellum (CER), and cerebral cortex (CC). In particular, neurotoxicity was evaluated at relatively low, medium, and high pesticide dosages. All pesticides inhibited acetylcholinesterase (AChE) and neuropathy target esterase (NTE) in each of the brain regions, but esterase inhibition was greatest in the HC and CS. Each of the pesticides also induced greater disruption to cellular bioenergetics within the HC and CS, and this was monitored via inhibition of mitochondrial complex enzymes I and II, reduced ATP levels, and increased lactate production. Similarly, the HC and CS were more vulnerable to redox stress, with greater inhibition of the antioxidant enzymes catalase and superoxide dismutase, and increased lipid peroxidation. All pesticides induced the production of nuclear Nrf2 in a dose-dependent manner. Collectively, these results suggest that the HC and CS are relatively vulnerable to pesticide exposures; brain regions that are typically damaged in certain NDDs such as Alzheimer’s disease and Parkinson’s disease.
General comment: This manuscript should be deeply reworked in order to improve its quality and impact. This version of the work seems to be unpublishable in this journal.
Some Major comments:
*) The main hypothesis of this work is not clear. Indeed, it is not clear whether the authors explore “the neurotoxicity of three widely utilized pesticides: malathion, chlorpyrifos, and paraquat, and their potential for differential damage to the hippocampus (HC), corpus striatum (CS), cerebellum (CER), and cerebral cortex (CC)”, or the authors explore a possible link “between excessive pesticide exposure and NDDs such as Alzheimer’s disease and Parkinson’s disease.” Although the authors state that “Hence, pesticide-induced damage that is focused on these regions could provide a basis for a molecular link between excessive pesticide exposure and these NDDs, but ultimately, this does not provide evidence of causal linkage, and only through more sophisticated and validated methods associated with accurate sampling and quantitation of acute or chronic exposures can a clearer link between pesticide exposures and NDDs be established”, it seems that the whole work is based on the exploration of a possible link “ between excessive pesticide exposure and NDDs such as Alzheimer’s disease and Parkinson’s disease”.
*) The main result of this work is not clear. Indeed the meaning of “Collectively, these results suggest that the HC and CS are relatively vulnerable to pesticide exposures” seems to be not clear. …. relatively to what ? Could the authors quantify something in this statement ?
*) The main aim should be narrowed only to the quantification of neurotoxic effects of pesticides in different brain regions, if novel.
*) All the experimental results should be presented in a clear way: the quality of plots should be improved to allow the interested readers to understand all the experimental achievements without the need of a time consuming exploration of the main text.
*) The quality of all figure captions should be improved in order to provide all the needed information directly within the captions and not only scattered within the main text.
*) Figures should be better organized in more clear panels. Perhaps 3,4 panels could be enough. Please rework accordingly.
*) All the sections “Results”, “Discussion” and “Conclusion” should be reworked. In particular, the authors write that: “This section may be divided by subheadings. It should provide a concise and precise description of the experimental results, their interpretation, as well as the experimental conclusions that can be drawn. “. Nevertheless, within the “Results” section only the experimental results should be presented without any comment. Comments and interpretations, as well as conclusions, should be moved to the “Discussion” and “Conclusion” sections, where the authors should discuss the value of their work with respect to the current state of the art.
Some minor comments:
*) Please check the quality and the labels on the axes of each figure
*) Please check the number of subsections 2.2 (2 subsection with the same number).
*) Legends and labels of each plots should be made readable.
Author Response
Reviewer #2.
Some Major comments:
*) The main hypothesis of this work is not clear. Indeed, it is not clear whether the authors explore “the neurotoxicity of three widely utilized pesticides: malathion, chlorpyrifos, and paraquat, and their potential for differential damage to the hippocampus (HC), corpus striatum (CS), cerebellum (CER), and cerebral cortex (CC)”, or the authors explore a possible link “between excessive pesticide exposure and NDDs such as Alzheimer’s disease and Parkinson’s disease.” Although the authors state that “Hence, pesticide-induced damage that is focused on these regions could provide a basis for a molecular link between excessive pesticide exposure and these NDDs, but ultimately, this does not provide evidence of causal linkage, and only through more sophisticated and validated methods associated with accurate sampling and quantitation of acute or chronic exposures can a clearer link between pesticide exposures and NDDs be established”, it seems that the whole work is based on the exploration of a possible link “ between excessive pesticide exposure and NDDs such as Alzheimer’s disease and Parkinson’s disease”.
*) The main result of this work is not clear. Indeed the meaning of “Collectively, these results suggest that the HC and CS are relatively vulnerable to pesticide exposures” seems to be not clear. …. relatively to what ? Could the authors quantify something in this statement ?
We apologise that we should have made the focus of the manuscript clear. The aim of the study was to consider the neurotoxicity of MAL, CPF, and PQ and to examine if there were differential effects between these pesticides on cellular bioenergetics and oxidative stress within different brain regions. Hence, in the revised manuscript we have amended the text of the abstract and introduction to make the aim of the work clear.
*) The main aim should be narrowed only to the quantification of neurotoxic effects of pesticides in different brain regions, if novel.
We agree that we could have made the aim of the manuscript more focused. The intention was to consider the neurotoxic effects of the pesticides and with a focus on their disruption of cellular bioenergetics. We have therefore amended the revised manuscript title and content, including the discussion, to focus on these aspects of the work.
*) All the experimental results should be presented in a clear way: the quality of plots should be improved to allow the interested readers to understand all the experimental achievements without the need of a time consuming exploration of the main text.
We have made the Figures larger and the key to the columns larger to assist the reader with rapidly identifying which of the columns relates to low, medium, or high pesticide dosing, and included further details in the Figure Legends.
*) The quality of all figure captions should be improved in order to provide all the needed information directly within the captions and not only scattered within the main text.
As detailed above, we have edited the Figure Legends to include more details of the experiments performed including details that the pesticides were administered at low, medium, and high dosages.
*) Figures should be better organized in more clear panels. Perhaps 3,4 panels could be enough. Please rework accordingly.
We have increased the size of each of the Figures as sometimes corruption of the embedded figures results in a lack of clarity of the image. In the finalized manuscript, high-resolution images will be supplied to the journal, and these will be embedded at a maximum size allowable by the journal editing team.
*) All the sections “Results”, “Discussion” and “Conclusion” should be reworked. In particular, the authors write that: “This section may be divided by subheadings. It should provide a concise and precise description of the experimental results, their interpretation, as well as the experimental conclusions that can be drawn. “. Nevertheless, within the “Results” section only the experimental results should be presented without any comment. Comments and interpretations, as well as conclusions, should be moved to the “Discussion” and “Conclusion” sections, where the authors should discuss the value of their work with respect to the current state of the art.
We appreciate this suggestion by Reviewer #2 and have deleted additional commentary text from the results section. We have also made amendments to the discussion section to make clear the aim of the study and the summary of the results.
Some minor comments:
*) Please check the quality and the labels on the axes of each figure
The labels have been checked and are listed as correct with additional details added to the Figure Legends.
*) Please check the number of subsections 2.2 (2 subsection with the same number).
Thank you for pointing this out, we have corrected the numbering of each of the subsections.
*) Legends and labels of each plots should be made readable.
The Figure Legends and axis labels for each of the plots have been checked and should all be legible.
Round 2
Reviewer 1 Report
I have received the revised version of the manuscript entitled "An investigation of the neurotoxic effects of malathion, chlorpyrifos, and paraquat to different brain regions." by Elmorsy et al. for re-evaluation. The authors have addressed all the comments satisfactorily. However, I have a couple of minor comments for the authors:
Minor comments:
1. In the animal protocols section authors have mentioned that they have decided on the doses of pesticide based on previous experiments performed in their laboratory. I would suggest citing the reference (if the data is already published) instead of writing “result not included”.
2. Authors should also mention how they measured the toxicity of pesticides (toxicity markers).
Author Response
We thank Reviewer 1 for reviewing the revised manuscript and providing further useful comments to improve the content and readability. We have addressed these comments below:
- In the animal protocols section authors have mentioned that they have decided on the doses of pesticide based on previous experiments performed in their laboratory.I would suggest citing the reference (if the data is already published) instead of writing “result not included”.
We appreciate the suggestion from Reviewer #1, however, the data has not yet been published for the intramuscular administration of the pesticides. However, we have undertaken other experimental work with these pesticides after dosing in mice, hens and rats and these references are included in the manuscript (references [4,25,26,58] of the revised manuscript). To provide the reader with more insight into the level of dosing, we have included three new references to the revised manuscript that detail the published pesticide LD50 values by several routes of administration (references 41-43 of the revised manuscript). Ultimately, toxicity monitoring is undertaken via animal observation (see below) and dosing for the OPs is also monitored via the level of cholinesterase inhibition, and this has been reported in Figures 1A-C.
- Authors should also mention how they measured the toxicity of pesticides (toxicity markers).
We appreciate that further details of the monitoring of dosed animals for signs of toxicity could have been included. We have therefore added these details into this experimental section of the revised manuscript. Copied as follows for convenience: Rats were monitored for signs of acute toxicity including drowsiness, lethargy and locomotor changes, as well as production of diarrhea, and changes in pupil size, food intake and/or body weight.
Reviewer 2 Report
Title : “The hippocampal and corpus striatum brain regions are relatively vulnerable to pesticide-induced serine hydrolase inhibition, bioenergetic disruption and damage from redox stress. “
In this study the authors explored the neurotoxicity of three widely utilized pesticides: malathion, chlorpyrifos, and paraquat, and their potential for differential damage to the hippocampus (HC), corpus striatum (CS), cerebellum (CER), and cerebral cortex (CC). In particular, neurotoxicity was evaluated at relatively low, medium, and high pesticide dosages. All pesticides inhibited acetylcholinesterase (AChE) and neuropathy target esterase (NTE) in each of the brain regions, but esterase inhibition was greatest in the HC and CS. Each of the pesticides also induced greater disruption to cellular bioenergetics within the HC and CS, and this was monitored via inhibition of mitochondrial complex enzymes I and II, reduced ATP levels, and increased lactate production. Similarly, the HC and CS were more vulnerable to redox stress, with greater inhibition of the antioxidant enzymes catalase and superoxide dismutase, and increased lipid peroxidation. All pesticides induced the production of nuclear Nrf2 in a dose-dependent manner. Collectively, these results suggest that the HC and CS are relatively vulnerable to pesticide exposures; brain regions that are typically damaged in certain NDDs such as Alzheimer’s disease and Parkinson’s disease.
General comment: Although the authors partially revised their manuscript some parts should be still reworked in order to enhance the quality and impact of the whole work. In particular, some sections are still mixed and should be reworked according to the standard of scientific contributions. In addition, the figures are still suboptimal and should be better organized in panels to help the interested readers to better understand the value of this work.
Some major comments are listed below:
*) Figures should be better organized in more clear panels. Perhaps 3,4 panels could be enough. Please rework accordingly.
*) All the sections “Results”, “Discussion” and “Conclusion” should be reworked. In particular, the authors write that: “This section may be divided by subheadings. It should provide a concise and precise description of the experimental results, their interpretation, as well as the experimental conclusions that can be drawn. “. Nevertheless, within the “Results” section only the experimental results should be presented without any comment. Comments and interpretations, as well as conclusions, should be moved to the “Discussion” and “Conclusion” sections, where the authors should discuss the value of their work with respect to the current state of the art.
Minor issues:
*) Some parts of the current version of the manuscript should be properly filled (e.g., Supplementary Materials, Author Contributions, Funding, Institutional Review Board Statement,
Informed Consent Statement, Data Availability Statement:, Acknowledgments: , Conflicts of Interest:,Appendix A , Appendix B.)
and lines :”
References ,
References must be numbered in order of appearance in the text (including citations in tables and legends) and listed individ- 631
ually at the end of the manuscript. We recommend preparing the references with a bibliography software package, such as 632
EndNote, ReferenceManager or Zotero to avoid typing mistakes and duplicated references. Include the digital object identifier 633
(DOI) for all references where available. 634
635
Citations and references in the Supplementary Materials are permitted provided that they also appear in the reference list here. 636
637
In the text, reference numbers should be placed in square brackets [ ] and placed before the punctuation; for example [1], [1–3] 638
or [1,3]. For embedded citations in the text with pagination, use both parentheses and brackets to indicate the reference number 639
and page numbers; for example [5] (p. 10), or [6] (pp. 101–105).”
Author Response
General comment: Although the authors partially revised their manuscript some parts should be still reworked in order to enhance the quality and impact of the whole work. In particular, some sections are still mixed and should be reworked according to the standard of scientific contributions. In addition, the figures are still suboptimal and should be better organized in panels to help the interested readers to better understand the value of this work.
Some major comments are listed below:
*) Figures should be better organized in more clear panels. Perhaps 3,4 panels could be enough. Please rework accordingly.
We do not concur with this comment. The work is focused on the cellular disruption of bioenergetics in response to pesticide treatment. An assessment of bioenergetics is an area of our combined expertise and we have published a number of papers in this area, for example:
Elmorsy et al 2021a. Antipsychotics inhibit the mitochondrial bioenergetics of pancreatic beta cells isolated from CD1 mice. Basic Clin Pharmacol Toxicol. 2021 Jan;128(1):154-168. doi: 10.1111/bcpt.13484.
Elmorsy et al 2021b. Effects of environmental metals on mitochondrial bioenergetics of the CD-1 mice pancreatic beta-cells. Toxicol In Vitro. 2021 Feb;70:105015. doi: 10.1016/j.tiv.2020.105015.
Alelwani et al 2020. Carbamazepine induces a bioenergetics disruption to microvascular endothelial cells from the blood-brain barrier. Toxicol Lett. 2020 Oct 15;333:184-191. doi: 10.1016/j.toxlet.2020.08.006.
Elmorsy et al 2017a. Adverse effects of anti-tuberculosis drugs on HepG2 cell bioenergetics. Hum Exp Toxicol. 2017 Jun;36(6):616-625. doi: 10.1177/0960327116660751.
Elmorsy et al 2017b. The role of oxidative stress in antipsychotics induced ovarian toxicity. Toxicol In Vitro. 2017 Oct;44:190-195. doi: 10.1016/j.tiv.2017.07.008.
Elmorsy et al 2017c. Effect of antipsychotics on mitochondrial bioenergetics of rat ovarian theca cells. Toxicol Lett. 2017 Apr 15;272:94-100. doi: 10.1016/j.toxlet.2017.03.018.
To thoroughly evaluate cellular bioenergetics, there is a need to consider several key parameters such as the activities of electron transport proteins (mitochondrial complexes), ATP and lactate levels, catalase and superoxide dismutase activities, and other markers of impact on oxidative stress such as levels of TBARS and Nrf2 activation. This is in accordance with our previously published data (sample publications detailed above) and therefore our results have logically been divided to cover these key measurements. To remove content entirely or to complicate it further by merging data would not be helpful to the reader.
*) All the sections “Results”, “Discussion” and “Conclusion” should be reworked. In particular, the authors write that: “This section may be divided by subheadings. It should provide a concise and precise description of the experimental results, their interpretation, as well as the experimental conclusions that can be drawn. “. Nevertheless, within the “Results” section only the experimental results should be presented without any comment. Comments and interpretations, as well as conclusions, should be moved to the “Discussion” and “Conclusion” sections, where the authors should discuss the value of their work with respect to the current state of the art.
We are confused with this comment as the wording that begins “this section may be divided….” is wording provided by the journal and not the manuscript authors. This text has been removed and substituted with the written content of the manuscript. Secondly, only wording that is directly related to the results has been included in the results section. The manuscript does not have a conclusion section to ‘reword’. Lastly, the discussion provides context to the work by considering other relevant publications in the field. The suggestion to simply rework this is not helpful or needed. Regarding reviewing, the journal recommendations suggest including specific comments (where required) and therefore the use of line numbers.
Minor issues:
*) Some parts of the current version of the manuscript should be properly filled (e.g., Supplementary Materials, Author Contributions, Funding, Institutional Review Board Statement,
Informed Consent Statement, Data Availability Statement:, Acknowledgments: , Conflicts of Interest:,Appendix A , Appendix B.)
and lines :”
These parts of the manuscript are either completed at submission (some were but do not show on the manuscript) and some are considered after acceptance of the manuscript. These details are added in consultation with the editorial team, they are not relevant for inclusion in the manuscript at this stage.